# SCALE-DISTRIBUTION DECOUPLING: ENABLING STABLE AND EFFECTIVE TRAINING OF LARGE LANGUAGE MODELS

## ABSTRACT

Training stability is a critical challenge in the pre-training of large language models (LLMs), particularly for architectures like Post-Norm Transformers prone to gradient explosion and dissipation. In this paper, we introduce Scale-Distribution Decoupling (SDD), a novel approach designed to enhance training stability by explicitly decoupling the scale and distribution of the weight matrix within fully-connected layers. SDD employs a normalization mechanism to regulate activation magnitudes and a learnable scaling vector to maintain well-conditioned gradients, thereby effectively preventing ***gradient explosion and dissipation*** and ensuring stable gradient propagation. This principled separation improves optimization efficiency, especially in deep networks. Extensive experiments across various LLM architectures (dense and MoE) demonstrate that SDD consistently achieves faster convergence and superior performance compared to existing normalization techniques. Furthermore, SDD is lightweight and seamlessly compatible with current frameworks, offering a practical and effective solution for robust LLM training.

## 1 INTRODUCTION

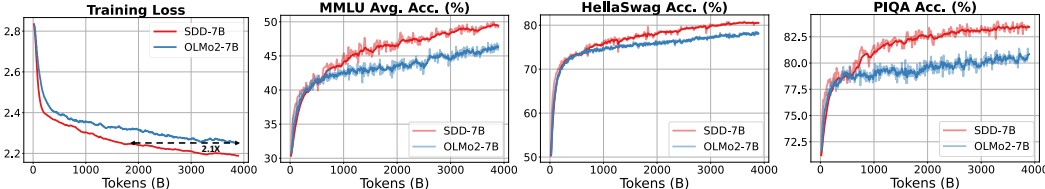

Figure 1: Training/validation loss with downstream performance on MMLU, HellaSwag and PIQA for 7B dense models trained with 4T tokens: SDD-7B (Post-Norm) achieves superior convergence (2.1×) and generalization over OLMo2-7B (Pre-Norm).

Large Language Models (LLMs) have demonstrated remarkable success in various natural language processing tasks (Li et al., 2024b; Zhu et al., 2024; Huang et al., 2025), fueled by advances in model architectures, large-scale datasets, and computational resources. However, the training stability of LLMs remains a critical challenge, especially as model size and complexity continue to grow. By "training stability", we refer to the ability of models to converge robustly without suffering from issues such as significant loss spikes, gradient explosion, or vanishing gradients, which often lead to optimization stagnation or divergence. Although Pre-Norm Transformer (Xiong et al., 2020; Zhuo et al., 2025) architectures exhibit greater stability during training, they are susceptible to feature collapse (Wang et al., 2024a; Xie et al., 2023), where representations across different layers become increasingly similar as depth increases, potentially hindering scalability. Post-Norm configurations, despite often yielding better final performance, remain significantly more difficult to train due to severe gradient explosion or vanishing issues, making stable optimization in such settings a persistent research challenge in LLM development (Zeng et al., 2022).

A fundamental source of these optimization challenges (Salimans & Kingma, 2016) lies in the inherent difficulty of effectively controlling the properties of weight matrices, particularly in deep,

high-dimensional networks. As models scale, the magnitude and distribution of weight parameters become increasingly coupled and difficult to manage during the optimization process. Although existing strategies, such as sophisticated initialization schemes (Zhang et al., 2019) and various normalization techniques applied to activations or weights (Ding et al., 2021; Xiong et al., 2020), offer valuable partial mitigation, they often do not directly address this core underlying issue: the tight entanglement between the weight matrix's overall scale and its internal directional components (distribution). This coupling forces gradient updates to simultaneously adjust both the magnitude and directional properties of weights, creating complex and volatile loss landscapes that contribute significantly to gradient instabilities like explosion or vanishing, slow convergence, and increased sensitivity to hyperparameters.

To tackle these challenges, we introduce **Scale-Distribution Decoupling (SDD)**, a novel and principled approach that fundamentally restructures fully-connected layers. At its core, SDD explicitly separates the optimization of weight matrix scale and distribution. This is achieved by applying a normalization step to standardize input activations, ensuring the subsequent weight transformation primarily focuses on learning the desired distribution, while a simple, learnable scaling vector is introduced to precisely control the overall magnitude of the layer's output activation. This direct decoupling mechanism contrasts with conventional layer formulations and existing normalization techniques that typically address activations or implicitly influence weight properties. By isolating scale and distribution optimization, this explicit strategy leads to more stable gradient propagation and simplified optimization dynamics. As a result, SDD substantially enhances training efficiency and stability, vividly illustrated by the significant performance gains and **2.1×** faster convergence observed in the 7B dense model training results shown in Figure 1.

SDD is lightweight, requires minimal architectural modifications, introducing only a small number of additional parameters (a scaling vector per layer), and seamlessly integrates with a wide range of model configurations. Empirical evaluations demonstrate that SDD consistently improves training stability across diverse LLM architectures, including notoriously unstable Post-Norm Transformers, and also enhances resistance to feature collapse in Pre-Norm models (as discussed further in Section 4). Furthermore, SDD accelerates convergence, improves generalization, and enables more efficient large-scale pre-training, making it a practical and effective solution for developing robust LLMs. The main contributions of this paper can be summarized as follows:

1. We introduce Scale-Distribution Decoupling (SDD), a novel and principled design that explicitly separates the scale and distribution of weight matrices. This approach addresses a fundamental limitation in LLM optimization by simplifying the learning dynamics and enhancing gradient stability.

2. We empirically demonstrate that SDD significantly stabilizes training across diverse LLM architectures, including both Pre-Norm and Post-Norm, effectively mitigating issues such as *gradient explosion, dissipation, and improving resistance to feature collapse*.

3. We provide extensive empirical evidence showing that our method consistently improves both convergence speed and training efficiency, illustrated by achieving similar training loss levels approximately **2.1×** faster in 7B models. This makes SDD highly applicable to large-scale pre-training tasks, and ultimately leads to superior downstream performance.

## 2 SCALE-DISTRIBUTION DECOUPLING

### 2.1 MOTIVATION

The training stability of large language models (LLMs) is frequently undermined by the challenges of optimizing high-dimensional weight matrices. Specifically, the scale of weight parameters has a profound impact on model outputs and gradient magnitudes, but is inherently difficult to learn effectively (Rybakov et al., 2024). Existing techniques, such as advanced initialization schemes and normalization strategies, provide partial mitigation but fail to address a fundamental issue: the entanglement of the weight matrix's scale and distribution. This entanglement introduces unnecessary complexity to the optimization process, especially in Post-Norm transformers, which are more susceptible to instability.

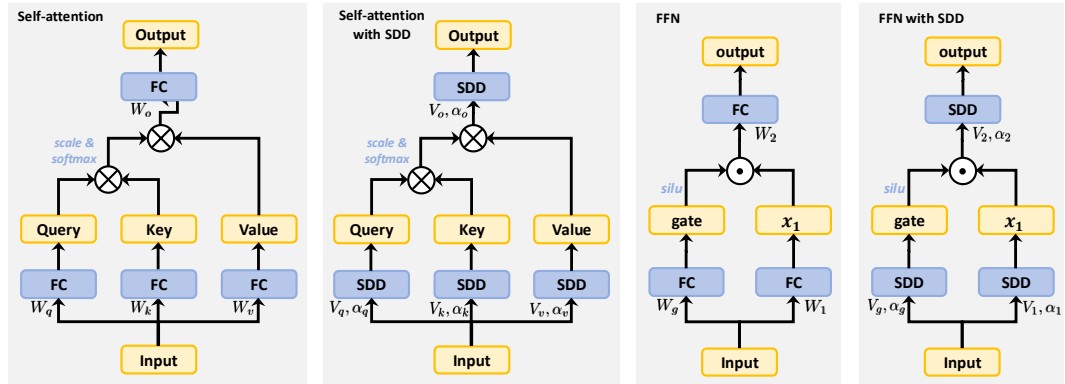

Figure 2: Comparison of vanilla and SDD-based Self-Attention and FFN Architectures. From left to right: the standard self-attention module, the self-attention module with SDD, the standard feed-forward network (FFN), and the SDD-based FFN. In these figures, "FC" represents a standard fully-connected layer, and "SDD" denotes the proposed Scale-Distribution Decoupled fully-connected layer, formally defined in Eqn. 2. Labels beneath "FC" and "SDD" indicate their learnable parameters. Notably, the additional parameter $\alpha$ in "SDD" is a one-dimensional vector per layer, contributing negligible overhead.

To address this issue, we propose **Scale-Distribution Decoupling (SDD)**, which disentangles the scale and distribution of weights in fully-connected layers. By isolating these two components, SDD not only simplifies the learning dynamics but also notably improves the training stability.

## 2.2 METHOD

In conventional fully-connected layers, the output is computed as:

$$y = Wx, \tag{1}$$

where $W \in \mathbb{R}^{n \times n}$ represents the learnable weight matrix and $x \in \mathbb{R}^n$ is the input vector. The SDD formulation modifies this operation to explicitly decouple the learning of scale and distribution:

$$y = \alpha \odot \mathrm{norm}(Vx), \tag{2}$$

where $V \in \mathbb{R}^{n \times n}$ is a learnable weight matrix responsible for the primary transformation, and $\odot$ denotes the element-wise multiplication, defined for vectors $a, b \in \mathbb{R}^n$ as $(a \odot b)_i = a_i b_i$. $\mathrm{norm}(\cdot)$ is a normalization function applied to the vector $z = Vx$ that removes the scale information while preserving its distribution. Following the normalization commonly used in Layer Normalization (LN) (Ba, 2016; Wang et al., 2022), we use Root Mean Square (RMS) normalization, defined as:

$$\mathrm{norm}(z) = \frac{z}{\|z\|_{RMS}}, \quad \text{where} \quad \|z\|_{RMS} = \sqrt{\frac{1}{n} \sum_{i=1}^{n} z_i^2}. \tag{3}$$

$\alpha \in \mathbb{R}^n$ is a learnable scaling vector to stabilize training during early stages and control the output magnitude (Figure 2).

This reformulation explicitly separates the roles of the weight parameters: $V$ is dedicated to learning the directional transformation, while $\alpha$ independently governs the output scale. Such a decoupling has two key advantages. First, it simplifies optimization by disentangling scale and distribution, reducing complex parameter interactions that hinder learning. Second, the combination of normalization and the learned scaling vector ensures controlled and bounded outputs, which inherently helps mitigate gradient-related issues such as explosion or vanishing. These properties make SDD particularly effective for training deep and wide models, improving stability in training.

SDD introduces minimal computational and memory overhead compared to standard fully-connected layers. The additional FLOPs for SDD per layer are approximately $6BSH$, where $B$ is the batch size, $S$ is the sequence length, and $H$ is the hidden size (assuming input/output dimensions are $H$). This accounts for about $3BSH$ for the RMS normalization and $3BSH$ for the

element-wise multiplication. Compared to the $O(BSH^2)$ FLOPs of the matrix multiplication $Vx$, this overhead is negligible, approximately $6/H$ for the layer. Across all feed-forward layers in a Transformer, this contributes only about $3/H$ of the total model FLOPs. The parameter overhead is similarly negligible, adding just $n$ (or $H$) parameters from the scaling vector $\alpha$ per layer, contributing $O(H)$ compared to $O(H^2)$ in the $V$ matrix, or $O(L \cdot H)$ globally where $L$ is the number of layers, a tiny fraction of the total model parameters. Given that $H > 1024$ in typical settings, both FLOPs and parameter overheads are negligible ($< 0.29\%$). Furthermore, SDD's additional memory cost during training can be effectively eliminated through gradient checkpointing.

## 3 THEORETICAL ANALYSIS

The SDD method is supported by a theoretical foundation that demonstrates its validity and advantages under common assumptions. To begin, we show that the proposed decoupling is equivalent to the standard fully-connected operation under Gaussian assumptions.

### 3.1 EXPRESSIVENESS OF STANDARD AND SDD-BASED LAYERS

Let $x \in \mathbb{R}^n$ be sampled from a standard Gaussian distribution $\mathcal{N}(0, I)$, and each element of $W \in \mathbb{R}^{n \times n}$ be i.i.d. Gaussian random variables with mean 0 and variance $\sigma^2/n$. For any fully-connected layer $y = Wx$, there exists an approximate representation $y = \alpha \odot \mathrm{norm}(Vx)$, where $\alpha \in \mathbb{R}^n$ is a vector and $V \in \mathbb{R}^{n \times n}$ is an matrix derived from $W$. Conversely, any output of the form $y = \alpha \odot \mathrm{norm}(Vx)$ can be approximately represented in the form $y = Wx$.

Its proof, demonstrating the approximate expressiveness between standard and SDD-based layers, is provided in Appendix D. *The expectation symbol $\mathbb{E}$ is omitted for brevity.* This equivalence encapsulates the fundamental principle of Scale-Distribution Decoupling (SDD): disentangling the scale and distribution of the weight matrix $W$. SDD achieves this by introducing a learnable scaling vector $\alpha$ to regulate magnitude, while $\mathrm{norm}(Vx)$ preserves the distributional structure of the transformed input. By explicitly decoupling these components, SDD streamlines optimization, obviating the need to simultaneously learn both scale and distribution. This separation enhances numerical stability, as $\alpha$ facilitates precise control over output magnitudes, while normalization ensures a well-conditioned distribution. Furthermore, SDD exhibits strong adaptability, seamlessly accommodating both orthogonal and general weight matrices $V$, making it a versatile and robust solution across diverse neural architectures.

### 3.2 GRADIENT ANALYSIS: STANDARD VS. SDD LAYERS

The gradients with respect to $\alpha$, $V$, and $x$ in the SDD-based formulation $y = \alpha \odot \mathrm{norm}(Vx)$ differ significantly from those in the standard fully-connected layer $y = Wx$:

1. The gradient with respect to $\alpha$ is well-conditioned and bounded, enabling faster and more stable optimization of the scale parameter.

2. The gradient with respect to $V$ is constrained by the normalization operation, ensuring bounded updates and avoiding gradient explosion or vanishing.

3. The gradient norm with respect to $x$ is moderated by the normalization operation, preventing gradient explosion or vanishing.

The key gradient formulas for standard and SDD layers are summarized below. Let $z = Vx$.

**Standard Fully-Connected Layer** ($y = Wx$)**:** The gradient with respect to the weight matrix $W$ is sensitive to the scales of $W$ and $x$, potentially leading to instability:

$$\frac{\partial \mathcal{L}}{\partial W} = \frac{\partial \mathcal{L}}{\partial y} x^\top, \quad \frac{\partial \mathcal{L}}{\partial x} = W^\top \frac{\partial \mathcal{L}}{\partial y}. \tag{4}$$

Their magnitudes are highly sensitive to the scale and condition of $W$ and the scale of $x$, often causing gradient instability (explosion or vanishing) in deep networks.

**SDD-Based Layer** ($y = \alpha \odot \mathrm{norm}(Vx)$)**:** *Gradient with Respect to $\alpha$:* The gradient for the learnable scaling vector $\alpha$ is well-conditioned due to the boundedness of $\mathrm{norm}(Vx)$:

$$\frac{\partial \mathcal{L}}{\partial \alpha} = \frac{\partial \mathcal{L}}{\partial y} \odot \mathrm{norm}(Vx). \tag{5}$$

*Approximate Gradient with Respect to $V$:* The approximate gradient with respect to the transformation matrix $V$ is given by a form involving the Frobenius norm of $V$, $\|V\|_F = \sqrt{\sum_{i,j} V_{i,j}^2}$. This form is influenced by the normalization operation, aiming to facilitate bounded updates for $V$:

$$\frac{\partial \mathcal{L}}{\partial V} \approx \frac{\alpha}{\|V\|_F} \odot \frac{\partial \mathcal{L}}{\partial y} \cdot \left( I - \frac{zz^\top}{n\|z\|_{RMS}^2} \right) \cdot \frac{x^\top}{\|x\|_{RMS}}. \tag{6}$$

*Approximate Gradient with Respect to $x$:* The approximate gradient propagated backward through the layer with respect to the input $x$ is moderated by the normalization operation, hypothesized to contribute to stable gradient flow. Under certain conditions and approximations, the norm of this gradient is expected to be approximately preserved:

$$\frac{\partial \mathcal{L}}{\partial x} \approx \frac{\alpha}{\|x\|_{RMS}} \odot \frac{\partial \mathcal{L}}{\partial y} \cdot \left( I - \frac{zz^\top}{n\|z\|_{RMS}^2} \right) \frac{V}{\|V\|_F}. \tag{7}$$

This leads to the approximate gradient norm preservation property:

$$\|\frac{\partial \mathcal{L}}{\partial x}\|_{RMS} \approx \|\frac{\partial \mathcal{L}}{\partial y}\|_{RMS}. \tag{8}$$

The detailed derivations for these gradient formulas and a more extensive discussion on their properties, including how the learning process of $\alpha$ helps mitigate the risk of vanishing gradients, are provided in Appendix E.

SDD enhances training stability by disentangling the scale and distributional components of the weight matrix. By introducing normalization into all fully-connected layers, SDD ensures gradients remain bounded, mitigating gradient explosion and dissipation. The learnable scaling vector $\alpha$ independently controls the scale, while the normalized transformation $\mathrm{norm}(Vx)$ isolates the distribution, improving the conditioning of $V$. These properties simplify optimization, enabling more robust and efficient training, especially in architectures prone to instability such as Post-Norm Transformers or high-dimensional layers. By addressing core challenges in large-scale neural network training, SDD provides a versatile and effective framework for stability and scalability.

## 4 EXPERIMENTS

We evaluate SDD on both dense and MoE models, measuring training stability, convergence speed, and downstream performance. Our experiments include large-scale benchmarks, ablation studies, and robustness tests. Results show that SDD consistently improves training efficiency, mitigates instability, and outperforms existing normalization techniques across various architectures and tasks.

### 4.1 EXPERIMENTAL SETUP

**Backbones.** We evaluate SDD on three Transformer architectures: two dense models (1B and 7B parameters) and an MoE model with 588M active parameters (3.4B in total). All baseline models are based on the OLMo2 (OLMo et al., 2024) and OLMoE (Muennighoff et al., 2024) frameworks, which utilize a Pre-Norm configuration. 1B dense model follows the OLMo2-1B architecture with 16 layers, $d_{model} = 2048$, 32 attention heads, and Grouped-Query Attention (GQA, 8 groups). 7B dense model utilizes an architecture with 32 layers, $d_{model} = 4096$, 32 attention heads, and Grouped-Query Attention (GQA, 8 groups). MoE model is based on the OLMoE architecture with 32 layers, $d_{model} = 1024$, 16 attention heads, and 64 experts (with 8 experts active per token). For fair comparison, all models are trained from scratch. Architectural details are summarized in Table 3 and full configurations provided in Appendix F. All models are trained on the OLMoE Mix dataset (Muennighoff et al., 2024). We compare SDD against several established normalization baselines:

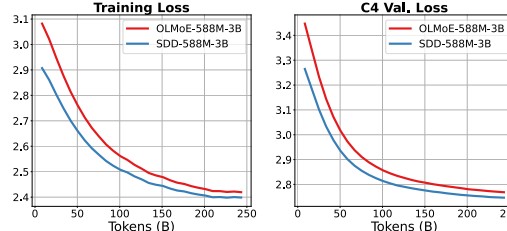

Figure 3: Training and validation loss on C4 for 1B dense models trained with 200B tokens.

Figure 4: Training and Validation Loss on C4 for MoE Models with 250B Tokens.

Table 1: Performance comparison of the 1B dense models. This table compares training loss, perplexity and downstream accuracy (%). "ARC-E" and "ARC-C" denote ARC-Easy and ARC-Challenge. The best results are in bold, and "Avg." represents average accuracy across tasks. SDD-1B achieves the best performance, demonstrating superior efficiency and generalization.

| Model | Loss ↓ | Perplexity ↓ | MMLU | HellaSwag | ARC-C | ARC-E | Winogrande | Openbook QA | COPA | Avg. ↑ |
|---|---|---|---|---|---|---|---|---|---|---|
| OLMo2-1B | 2.70 | 14.88 | 34.06 | 56.98 | 34.11 | 66.90 | 58.25 | 35.80 | 78.00 | 52.01 |
| PostNorm-1B | 2.69 | 14.73 | 32.94 | 57.78 | 32.66 | 65.96 | 58.22 | 37.33 | 79.33 | 52.03 |
| DeepNorm-1B | 2.72 | 15.18 | 33.06 | 55.73 | 31.77 | 65.09 | 55.99 | 35.67 | 79.67 | 51.00 |
| Mix-LN | 2.68 | 14.59 | 34.03 | 57.63 | 33.18 | 67.58 | 59.10 | 35.48 | 76.80 | 51.97 |
| nGPT-1B | 2.71 | 15.03 | 33.02 | 55.81 | 32.41 | 65.50 | 57.52 | 36.21 | 79.35 | 51.40 |
| SDD-1B | **2.65** | **14.15** | **34.71** | **59.65** | **37.57** | **69.65** | **59.06** | **37.33** | **80.33** | **54.04** |

Pre-Norm (the default OLMo/OLMoE configuration), Post-Norm (Vaswani et al., 2017), DeepNorm (Wang et al., 2024a) and nGPT (Loshchilov et al., 2024) .

**Training Setup.** All models are trained using the AdamW optimizer ($\beta_1 = 0.9, \beta_2 = 0.95$) on sequences of 4096 tokens. Initialization for baseline models follows the schemes used in OLMo2 (Groeneveld et al., 2024) and OLMoE (Muennighoff et al., 2024), combining truncated normal (Groeneveld et al., 2024) and DS-Init (Zhang et al., 2019). For SDD, the learnable scaling vector $\alpha$ is initialized specifically: $1/\sqrt{\text{number of layers}}$ for the output projection mappings in attention and feed-forward networks (FFNs), and 1 for other projection layers (query, key, value, gating). The remaining weight parameters ($V$ matrices) are initialized using a normal distribution $\mathcal{N}(0, 1/\sqrt{2.5 \cdot d_{model}})$, a strategy designed to align the initial output distributions with the baselines and promote stable training start. The learning rate schedule follows a cosine decay: the dense model uses a peak learning rate of $3e^{-4}$ decaying to $1.5e^{-5}$, while the MoE model uses $4e^{-4}$ decaying similarly. Training is conducted on NVIDIA H800 80GB GPUs with a global batch size of 1024 and a micro-batch size of 4 per device, optimizing the next-token prediction Negative Log Likelihood (NLL) loss. Gradient clipping (max norm 1.0) and BF16 mixed precision are used across all experiments to ensure stable and efficient training.

**Evaluation.** We evaluate SDD across benchmarks covering reasoning, commonsense understanding, and question answering. Reasoning tasks include ARC-Easy, ARC-Challenge (Clark et al., 2018), PIQA (Bisk et al., 2020), and MMLU (Hendrycks et al., 2021). Commonsense understanding is assessed via HellaSwag (Zellers et al., 2019), Winogrande (Sakaguchi et al., 2021), SocialIQA (Sap et al., 2019), and CSQA (Talmor et al., 2019). For question answering, we use SciQ (Welbl et al., 2017), CoQA (Reddy et al., 2019), BoolQ (Clark et al., 2019), COPA (Gordon et al., 2012), and OBQA (Mihaylov et al., 2018).

## 4.2 RESULTS ON DENSE MODEL

This section presents our experimental evaluation of SDD on dense Transformer models at both the 1B and 7B parameter scales. At the 1B scale, we compare SDD-1B (Post-Norm) against several normalization baselines, including OLMo2-1B (Pre-Norm), PostNorm-1B (Post-Norm), DeepNorm-1B (Post-Norm), Mix-LN (Li et al., 2024a), and nGPT-1B (Post-Norm). These models were trained up to 200 billion tokens, with longer runs for OLMo2-1B and SDD-1B extending to 2 trillion tokens. At the 7B scale, we evaluate SDD-7B (Post-Norm) against the OLMo2-7B (Pre-Norm) baseline trained on 4 trillion tokens. These experiments are designed to assess the impact of SDD on training stability, convergence speed, and downstream performance across different model sizes and training durations. Detailed results for the 1B and 7B models are presented in the following statement.

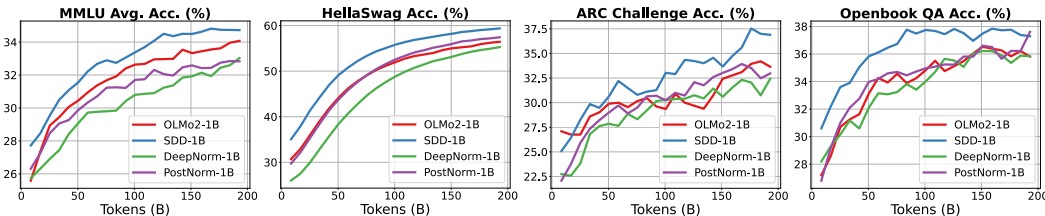

Figure 5: Downstream performance on MMLU, HellaSwag, ARC-Challenge, and OpenbookQA for dense models trained on 200B tokens. SDD-1B consistently outperforms others, showcasing superior generalization.

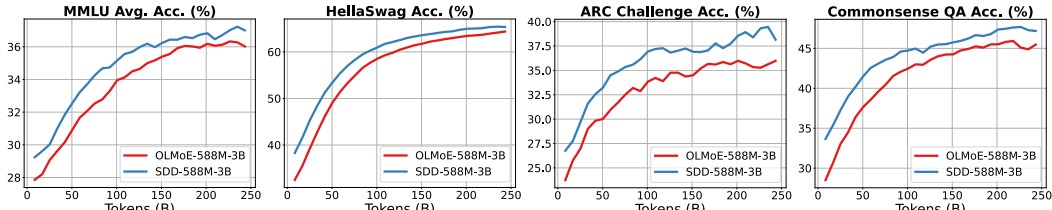

Figure 6: Downstream performance on MMLU, HellaSwag, ARC-Challenge, and Commonsense for MoE models with 250 billion training tokens.

**Training Dynamics of 1B Dense Model.** Figure 3 shows the training and validation loss on C4 for 1B dense models trained with 200B tokens. Among OLMo2-1B (Pre-Norm), PostNorm-1B, DeepNorm-1B (both Post-Norm), nGPT-1B (Post-Norm) and SDD-1B (Post-Norm), SDD-1B converges faster and reaches the lowest loss. It achieves 2.65, outperforming OLMo2-1B (2.70), PostNorm-1B (2.69), DeepNorm-1B (2.72) and nGPT-1B (2.71), demonstrating superior stability and efficiency. These highlight SDD's ability to improve optimization by decoupling scale and distribution.

**Downstream Evaluation of 1B Dense Model.** Table 1 and Figure 5 summarize downstream results across MMLU, HellaSwag, ARC-Challenge, ARC-Easy, Winogrande, Openbook QA, and COPA. SDD-1B consistently outperforms its counterparts, achieving the highest average accuracy of 54.04%, surpassing OLMo2-1B (52.01%), PostNorm-1B (52.03%), DeepNorm-1B (51.00%) and nGPT-1B (51.40). Notable gains include a 3.46% and 2.67% improvement over the second-best model on ARC-Challenge (37.57%) and HellaSwag (59.65%), respectively. These results reinforce SDD-1B's effectiveness in capturing complex linguistic patterns and improving generalization across diverse benchmarks.

**Scaling on 7B Dense Model.** We also evaluate SDD on a larger 7B dense model scale, comparing SDD-7B (Post-Norm) against the OLMo2-7B (Pre-Norm) baseline trained on 4T tokens, as shown in Figure 1. SDD-7B demonstrates significantly faster convergence in training loss, reaching a similar loss level approximately **2.1×** faster than the OLMo2-7B baseline. Furthermore, SDD-7B consistently achieves higher downstream performance on MMLU, HellaSwag, and PIQA throughout the training process. These results indicate that the benefits of SDD scale effectively to larger dense models and longer training durations.

### 4.3 RESULTS ON MOE MODEL

We evaluate SDD on OLMoE-588M-3B, an MoE model with 588M active parameters out of 3.4B total (Muennighoff et al., 2024). Due to computational constraints, we compare it to the baseline OLMoE-588M-3B with identical hyperparameters. SDD introduces only a 0.1% increase in parameters due to the scaling vector $\alpha$, ensuring a fair comparison without modifying training settings.

**Training dynamics of MoE model.** Figure 4 presents the training and validation loss curves for MoE models trained on 250B tokens. SDD-588M-3B consistently achieves lower losses than OLMoE-588M-3B, demonstrating improved convergence and stability. This suggests that SDD not only accelerates training but also mitigates optimization challenges common in large-scale MoE.

**Downstream Evaluation.** Figure 6 shows that SDD-588M-3B outperforms OLMoE-588M-3B across all benchmarks, particularly in MMLU, which evaluates multi-domain reasoning. More met-

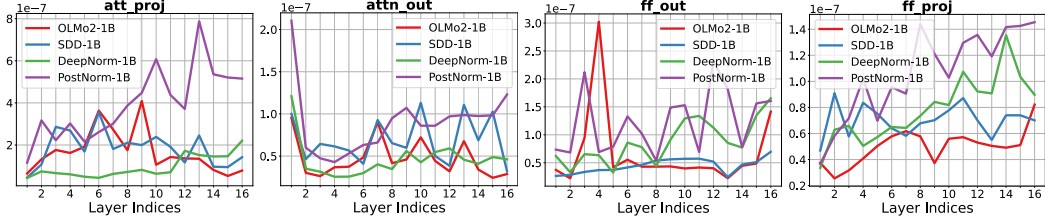

Figure 7: Comparison of Gradient Norms Across Layers. We compare four methods: OLMo2-1B (Pre-Norm), PostNorm-1B, DeepNorm-1B, and SDD-1B (all Post-Norm). "att_proj" refers to the query/key/value projection, "attn_out" to the attention output projection, "ff_proj" to the gating and first FC layer in the feed-forward network (FFN), and "ff_out" to the second FC layer in the FFN.

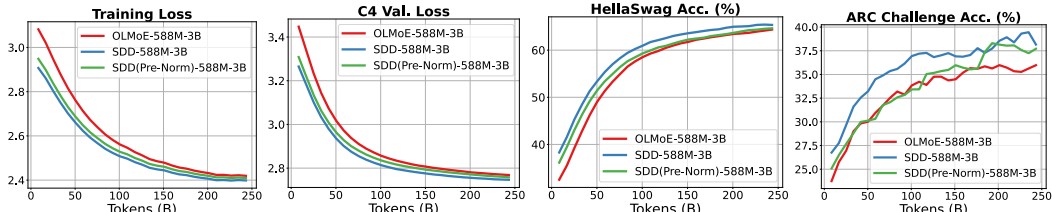

Figure 8: Training and downstream performance of SDD-588M-3B with Pre-Norm and Post-Norm compared to OLMoE-588M-3B (Pre-Norm). Models trained on 250 billion tokens show that SDD improves convergence speed and downstream accuracy in the Pre-Norm setting. Switching to Post-Norm with SDD yields even greater performance gains.

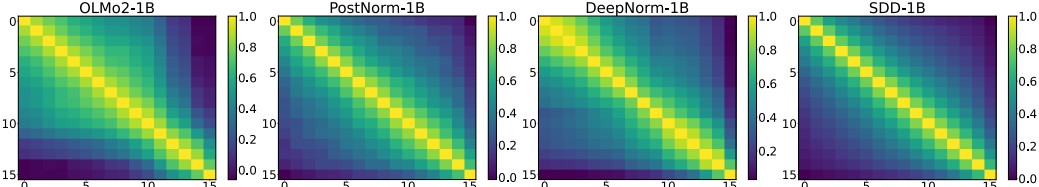

Figure 9: Layer-Wise Feature Similarity Across Normalization Methods. This figure compares feature similarity across layers in OLMo2-1B, PostNorm-1B, DeepNorm-1B, and SDD-1B. SDD-1B achieves the highest inter-layer similarity, indicating more stable feature propagation.

rics are available in Appendix H. These improvements underscore SDD's capacity to enhance generalization and capture intricate linguistic patterns. Overall, SDD boosts both training efficiency and downstream performance in MoE architectures, providing a robust and scalable solution for large-scale model optimization.

## 4.4 ABLATION STUDY

**Gradient Visualization.** Figure 7 compares gradient norms across layers for OLMo2-1B (Pre-Norm), PostNorm-1B, DeepNorm-1B (both Post-Norm), and SDD-1B (Post-Norm). SDD-1B maintains significantly more stable gradient norms, mitigating gradient explosion and vanishing, which commonly affect Post-Norm variants. This stability improves optimization and training robustness, especially in deep networks, making SDD particularly effective for large-scale models.

**SDD on Pre-Norm.** Figure 8 evaluates SDD-588M-3B under both Pre-Norm and Post-Norm settings. When applied to Pre-Norm, SDD accelerates convergence and enhances downstream accuracy. Further gains are observed when transitioning from Pre-Norm to Post-Norm, highlighting SDD's adaptability and effectiveness in improving training stability and generalization.

**Layer-wise Similarity.** Figure 9 illustrates inter-layer feature similarity across normalization methods. SDD-1B exhibits the lowest similarity, indicating reduced feature redundancy and effectively mitigating feature collapse. This suggests that SDD promotes more diverse representations across layers, contributing to better optimization and enhanced generalization.

Table 2: Impact of Hyperparameter Perturbations on Model Performance. "−" indicates non-convergence. All models are trained on 200B tokens. "lr∗5" refers to a 5x increase in learning rate, "Initstd∗0.1" scales the initialization standard deviation by 0.1, and "wo Warmup" denotes the removal of the warmup phase.

| Model | lr∗5 | Initstd∗0.1 | wo WarmUp | Loss ↓ | Perplexity ↓ |
|---|---|---|---|---|---|
| OLMo2-581M | ✗ | ✗ | ✗ | 2.85 | 17.29 |
| OLMo2-581M | ✓ | ✗ | ✗ | 2.84 | 17.12 |
| OLMo2-581M | ✗ | ✓ | ✗ | 2.86 | 17.46 |
| OLMo2-581M | ✗ | ✗ | ✓ | 2.85 | 17.29 |
| PostNorm-581M | ✗ | ✗ | ✗ | 2.84 | 17.12 |
| PostNorm-581M | ✓ | ✗ | ✗ | − | − |
| PostNorm-581M | ✗ | ✓ | ✗ | − | − |
| PostNorm-581M | ✗ | ✗ | ✓ | − | − |
| DeepNorm-581M | ✗ | ✗ | ✗ | 2.84 | 17.12 |
| DeepNorm-581M | ✓ | ✗ | ✗ | − | − |
| DeepNorm-581M | ✗ | ✓ | ✗ | − | − |
| DeepNorm-581M | ✗ | ✗ | ✓ | 2.87 | 17.64 |
| SDD-581M | ✗ | ✗ | ✗ | 2.83 | 16.95 |
| SDD-581M | ✓ | ✗ | ✗ | **2.81** | **16.61** |
| SDD-581M | ✗ | ✓ | ✗ | 2.82 | 16.78 |
| SDD-581M | ✗ | ✗ | ✓ | 2.83 | 16.95 |

**Robustness on Hyperparameter Perturbations.** Table 2 assesses model robustness under hyperparameter variations, including increased learning rates, reduced initialization scale, and removal of warmup. While PostNorm-581M and DeepNorm-581M fail to converge under certain conditions, SDD-581M consistently stabilizes training and achieves lower loss, demonstrating resilience to hyperparameter changes.

**Scaling law for model depth.** Figure 10 compares OLMo2-1B (Pre-Norm) and SDD-1B (Post-Norm) across varying depths. SDD enables deeper models to scale effectively, overcoming training instability that typically limits Post-Norm architectures. This is particularly evident as the depth increases, where SDD

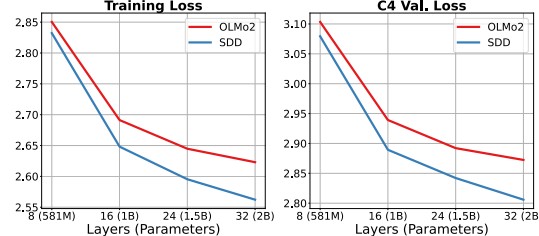

Figure 10: Scaling with model depth: OLMo2-1B (Pre-Norm) vs. SDD-1B (Post-Norm). All models are trained on 200 billion tokens, with only the number of layers varied. Notably, SDD-1B (Post-Norm) exhibits a clearly superior scaling law as the number of layers increases.

maintains stability and ensures smooth optimization. These results further validate SDD's ability to improve convergence and performance in large-scale Transformer models, making it a promising solution for very deep architectures.

## 5 CONCLUSION

We propose Scale-Distribution Decoupling (SDD), a method that stabilizes Transformer training by explicitly separating the scale and distribution of fully connected layer parameters. Our theoretical analysis establishes its expressivity and training benefits, while gradient analysis confirms improved stability, reducing the risk of gradient explosion or vanishing. Extensive experiments on both dense and MoE models demonstrate that SDD accelerates convergence, improves generalization, and enhances robustness to hyperparameter perturbations. It further delivers up to $2.1\times$ faster convergence and higher accuracy, with superior depth scaling validated across dense and MoE LLMs under budgets up to 200B–4T tokens. Additionally, SDD exhibits superior scalability with depth and fosters more consistent inter-layer representations. By addressing key training challenges, SDD provides a principled approach for improving the efficiency and stability of LLMs.

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

## A  RELATED WORK

**Normalization Techniques in Transformers.** Normalization is essential for stabilizing deep Transformer training (Wang et al., 2024b; 2022), with Layer Normalization (LN) (Ba, 2016; Wang et al., 2022) being the standard. Pre-Norm (Xiong et al., 2020) improves stability but often reduces expressivity, while Post-Norm (Vaswani et al., 2017) enhances generative performance but is prone to gradient explosion in deep networks. Approaches like DeepNorm (Wang et al., 2024a) and Sandwich-LN (Ding et al., 2021) aim to address these challenges by balancing stability and expressivity. Our method, Scale-Distribution Decoupling (SDD), builds on these efforts by explicitly disentangling the scale and distribution of the weight matrix, preserving stability while enhancing expressivity and optimizing training.

**Mixture of Experts and Large-Scale Model Training.** The adoption of Mixture of Experts (MoE) architectures (Shazeer et al., 2017; Fedus et al., 2022) has allowed for more efficient computation by activating subsets of parameters per forward pass. However, MoE introduces instability in expert selection and training divergence. OLMoE (Muennighoff et al., 2024) and architectures like Switch Transformers (Fedus et al., 2022) mitigate these issues with improved routing and load balancing. SDD complements these approaches by enhancing convergence and robustness, ensuring MoE models remain stable even under varying hyperparameter settings.

**Scaling and Stability in Large Language Models.** Training stability becomes more difficult as Transformer depth increases, with gradient-related issues like vanishing or exploding gradients. Techniques such as T-Fixup (Huang et al., 2020) and GradNorm (Chen et al., 2018) focus on balancing gradient magnitudes, while DS-Init (Zhang et al., 2019) improves initialization. However,

these methods primarily address stability from a weight-scaling perspective, rather than tackling optimization dynamics directly. SDD addresses these challenges by improving depth scalability and maintaining stable feature representations across layers, reducing redundancy, and mitigating feature collapse. These advantages make SDD a robust solution for training large-scale Transformers.

By addressing both stability and expressivity, SDD offers a scalable and efficient solution that enhances training stability while preserving the model's capacity to capture complex patterns. This decoupling of scale and distribution ensures robust optimization, enabling effective training of modern Transformer architectures, even in deep or high-dimensional networks, while maintaining model performance.

## B  BROADER IMPACTS AND LIMITATIONS

### B.1  BROADER IMPACTS

This paper proposes **Scale-Distribution Decoupling (SDD)**, a novel technique that improves the training stability, convergence speed, and performance of large language models (LLMs), particularly Transformer architectures. By addressing fundamental optimization challenges like gradient instability and enabling more robust training, SDD has the potential to assist the LLM research and development community in building and training more reliable, efficient, and capable models. This advancement could facilitate progress in developing more powerful AI systems for various beneficial applications. While, like any significant advancement in AI technology, our work may have broader societal implications, we do not identify any specific negative impacts inherent to SDD itself that must be particularly highlighted here, beyond those generally associated with the development and deployment of powerful LLMs.

### B.2  LIMITATIONS

Although our theoretical analysis demonstrates how SDD's decoupling mechanism can lead to improved gradient stability and simplified optimization dynamics under specific assumptions, these theoretical findings do not directly guarantee superior overall model performance or generalization on all complex downstream tasks compared to all possible alternative architectural modifications or normalization methods. Empirical evaluation remains the ultimate arbiter of overall effectiveness.

## C  DISCLOSURE OF LLM USAGE

In accordance with the ICLR 2026 policy on the disclosure of large language model (LLM) usage, we hereby state the following:

We used LLMs only to aid in language polishing and grammar refinement. All conceptualization, methodological design, experiments, and analyses were carried out solely by the authors.

## D  PROOF OF EXPRESSIVENESS EQUIVALENCE

*Proof. (1)* $y = Wx \implies y = \alpha \odot \mathrm{norm}(Vx)$.

Let $W \in \mathbb{R}^{n \times n}$ be the weight matrix of a fully-connected layer, where each element of $W$ is sampled from an independent Gaussian distribution $\mathcal{N}(0, \sigma^2/n)$. Using singular value decomposition (SVD), $W$ can be written as:

$$W = U\Sigma V'^{\top}, \tag{9}$$

where $U \in \mathbb{R}^{n \times n}$ and $V' \in \mathbb{R}^{n \times n}$ are orthogonal matrices, and $\Sigma \in \mathbb{R}^{n \times n}$ is a diagonal matrix containing the singular values $\sigma_1, \sigma_2, \ldots, \sigma_n$ of $W$. Substituting $W$ into $y = Wx$, we can rewrite the output as:

$$y = Wx = U\Sigma V'^{\top}x. \tag{10}$$

Let $z = V'^{\top}x$. Since $x \sim \mathcal{N}(0, I)$, the orthogonal transformation $V'^{\top}x$ preserves the Gaussian distribution of $x$, meaning $z \sim \mathcal{N}(0, I)$. According to Theorem 3.1.1 (Vershynin, 2018), $\|x\|_{RMS}$

is approximately equal to 1. So for simplicity, we set $\|z\|_{RMS} = 1$. The term $\Sigma z$ scales the components of $z$ along the singular directions, where:

$$\Sigma z = [\sigma_1 z_1, \sigma_2 z_2, \ldots, \sigma_n z_n]^\top, \tag{11}$$

The orthogonal matrix $U$ then rotates the scaled vector $\Sigma z$:

$$y = U\Sigma z. \tag{12}$$

Next, we normalize $y$, effectively removing the rotational effect of $U$:

$$\text{norm}(y) = \text{norm}(U\Sigma z) = \frac{U\Sigma z}{\|U\Sigma z\|_{RMS}} = \frac{U\Sigma z}{\|\Sigma z\|_{RMS}} = U \cdot \text{norm}(\Sigma z). \tag{13}$$

where $\|\Sigma z\|_{RMS}$ denotes the norm of the diagonal matrix $\Sigma z$. Thus, $U$ can always be absorbed into subsequent layers' mappings in a neural network without affecting the overall output, regardless of whether normalization is applied between $U$ and the subsequent layers. For example, in Transformer models, $U$ can propagate through the value, output projection of attention, and feed-forward network (FFN) mappings, making its explicit presence inconsequential. In other words, $y' = \Sigma V'^\top x$ is equivalent in expressiveness to $y = Wx$ in Transformer models. Therefore, we make no distinction between $y'$ and $y$.

After absorbing $U$, and noting that $\|z\|_{RMS} = 1$ implies $\text{norm}(z) = z$, the output can be reformulated as:

$$y = \alpha \odot V'^\top x = \alpha \odot \text{norm}(V'^\top x), \tag{14}$$

where $\alpha = \text{diag}(\Sigma) = [\sigma_1, \sigma_2, \ldots, \sigma_n]^\top$ captures the scale information of $W$. Let $V = V'^\top$, then the output $y = Wx$ can be equivalently expressed in the form $y = \alpha \odot \text{norm}(Vx)$.

*(2) $y = \alpha \odot \text{norm}(Vx) \implies y = Wx$.*

Consider the representation $y = \alpha \odot \text{norm}(Vx)$, where $\alpha \in \mathbb{R}^n$ is a vector, and $V \in \mathbb{R}^{n \times n}$ is a general matrix that may not necessarily be orthogonal. To demonstrate that this output can be equivalently expressed as $y = Wx$, the matrix $V$ is decomposed using singular value decomposition (SVD). Specifically, let:

$$V = P\Lambda Q^\top, \tag{15}$$

where $P \in \mathbb{R}^{n \times n}$ and $Q \in \mathbb{R}^{n \times n}$ are orthogonal matrices, and $\Lambda \in \mathbb{R}^{n \times n}$ is a diagonal matrix containing the singular values of $V$, denoted as $\gamma_1, \gamma_2, \cdots, \gamma_n$.

Substituting the decomposition of $V$ into the given equation, the output becomes:

$$y = \alpha \odot \text{norm}(Vx) = \alpha \odot \text{norm}(P\Lambda Q^\top x). \tag{16}$$

Define $z = Q^\top x$. Since $x \sim \mathcal{N}(0, I)$, and by Theorem 3.1.1 (Vershynin, 2018), $\|x\|_{RMS}$ is approximately 1. For brevity, we assume $\|x\|_{RMS} = 1$. The orthogonality of $Q$ guarantees that $\|z\|_{RMS} = 1$. Therefore, the expression for $y$ can now be written as:

$$y = \alpha \odot \text{norm}(P\Lambda z). \tag{17}$$

To simplify further, note that the normalization operation satisfies:

$$\text{norm}(P\Lambda z) = P \cdot \text{norm}(\Lambda z). \tag{18}$$

For a diagonal matrix $\Lambda$, the normalization of $\Lambda z$ can be approximately expressed as:

$$\text{norm}(\Lambda z) = \frac{\Lambda z}{\|\Lambda z\|_{RMS}} \approx \frac{\Lambda z}{\|\Lambda\|_{RMS}}. \tag{19}$$

Here, $\|\Lambda\|_{RMS} = \sqrt{(\gamma_1^2 + \gamma_2^2 + \cdots + \gamma_n^2)/n}$. Substituting this result, the output becomes:

$$y = \alpha \odot P\frac{\Lambda z}{\|\Lambda\|_{RMS}}. \tag{20}$$

By substituting back $z = Q^\top x$, we have:

$$y = \alpha \odot P\frac{\Lambda}{\|\Lambda\|_{RMS}}Q^\top x. \tag{21}$$

The equivalence to $y = Wx$ is now established by defining:

$$W = \alpha \odot P\frac{\Lambda}{\|\Lambda\|_{RMS}}Q^\top. \tag{22}$$

Thus, $W \in \mathbb{R}^{n \times n}$ is a valid weight matrix that satisfies $y = Wx$ for any $x$, completing the proof of reverse equivalence. $\qquad\square$

# E DETAILED GRADIENT DERIVATIONS

*Proof.* For the standard fully-connected layer $y = Wx$, the gradient with respect to $W$, which encodes both scale and distributional properties, is:

$$\frac{\partial \mathcal{L}}{\partial W} = \frac{\partial \mathcal{L}}{\partial y} \cdot x^\top, \tag{23}$$

where $\frac{\partial \mathcal{L}}{\partial y}$ is the backpropagated gradient. The magnitude of $\frac{\partial \mathcal{L}}{\partial W}$ is highly sensitive to the initialization of both $W$ and $x$. Poorly scaled $W$ or $x$ can lead to gradient explosion or vanishing, complicating optimization. In contrast, the SDD-based formulation $y = \alpha \odot \mathrm{norm}(Vx)$ decouples these components, leading to the following gradient properties:

*Gradient with Respect to $\alpha$:* The scale parameter $\alpha$, is explicitly learned in the SDD formulation, with its gradient given by:

$$\frac{\partial \mathcal{L}}{\partial \alpha} = \frac{\partial \mathcal{L}}{\partial y} \odot \mathrm{norm}(Vx). \tag{24}$$

Since $\mathrm{norm}(Vx)$ is bounded due to the normalization operation, $\frac{\partial \mathcal{L}}{\partial \alpha}$ remains stable and well-conditioned throughout training. Unlike the standard formulation, where scale and distribution are entangled in $W$, the decoupling in SDD allows $\alpha$ to be optimized independently. This results in consistently larger and more stable gradient updates for $\alpha$, enabling faster convergence of the scale parameter.

*Gradient with Respect to $V$:* The distributional characteristics of the input are controlled by $V$ in the SDD formulation. Given $z = Vx$, the gradient of the loss function $\mathcal{L}$ with respect to $V$ is expressed as:

$$\frac{\partial \mathcal{L}}{\partial V} = \frac{\partial \mathcal{L}}{\partial y} \cdot \frac{\partial y}{\partial V}. \tag{25}$$

Since $y = \alpha \odot \mathrm{norm}(z)$, we have:

$$\frac{\partial y}{\partial V} = \alpha \odot \frac{\partial \mathrm{norm}(z)}{\partial V}. \tag{26}$$

The chain rule gives:

$$\frac{\partial \mathrm{norm}(z)}{\partial V} = \frac{\partial \mathrm{norm}(z)}{\partial z} \cdot \frac{\partial z}{\partial V}. \tag{27}$$

Using the formula for the gradient of the normalized vector:

$$\frac{\partial \mathrm{norm}(z)}{\partial z} = \frac{1}{\|z\|_{RMS}} \left( I - \frac{zz^\top}{n\|z\|_{RMS}^2} \right), \tag{28}$$

and $\frac{\partial z}{\partial V} = x^\top$. Substituting this into the gradient of $\mathcal{L}$ with respect to $V$:

$$\frac{\partial \mathcal{L}}{\partial V} = \frac{\alpha}{\|z\|_{RMS}} \odot \frac{\partial \mathcal{L}}{\partial y} \cdot \left( I - \frac{zz^\top}{n\|z\|_{RMS}^2} \right) \cdot x^\top. \tag{29}$$

Next, assuming that $V$ and $x$ are i.i.d. with elements following a standard normal distribution $\mathcal{N}(0, \sigma^2)$, we further simplify the expression. Let $\|V\|_F$ denote the Frobenius norm of $V$, which is defined as:

$$\|V\|_F = \sqrt{\sum_{i,j} V_{i,j}^2}. \tag{30}$$

Incorporating this definition, the gradient becomes:

$$\frac{\partial \mathcal{L}}{\partial V} \approx \frac{\alpha}{\|V\|_F} \odot \frac{\partial \mathcal{L}}{\partial y} \cdot \left( I - \frac{zz^\top}{n\|z\|_{RMS}^2} \right) \cdot \frac{x^\top}{\|x\|_{RMS}}. \tag{31}$$

A key observation is that $\frac{\partial \mathcal{L}}{\partial y}$ remains stable across layers, with its magnitude exhibiting minimal fluctuations as it propagates through the network. This stability will be formally demonstrated in the subsequent gradient analysis with respect to $x$. Consequently, the gradient norm of $\frac{\partial \mathcal{L}}{\partial V}$ is primarily

determined by $\|V\|_F$, ensuring robustness during training. Furthermore, this stability enables precise control over $\frac{\partial \mathcal{L}}{\partial V}$ via adjusting the standard deviation (std) of $V$. By simply initializing $V$ with small values, we can enhance convergence speed and improve overall training efficiency.

*Gradient with Respect to $x$:* In the standard fully-connected layer, the gradient of the loss $\mathcal{L}$ toward the input $x$ is:

$$\frac{\partial \mathcal{L}}{\partial x} = W^\top \cdot \frac{\partial \mathcal{L}}{\partial y}. \tag{32}$$

The gradient depends entirely on the transpose of the weight matrix $W$ and the backpropagated gradient $\frac{\partial \mathcal{L}}{\partial y}$. In this formulation, the gradient magnitude is sensitive to the scale and condition of $W$, meaning poorly scaled or ill-conditioned weight matrices can lead to gradient explosion or dissipation. Large singular values in $W$ amplify the gradient norm, resulting in unstable optimization due to gradient explosion, while small singular values reduce the gradient norm, leading to gradient dissipation and slowed convergence.

The SDD formulation $y = \alpha \odot \mathrm{norm}(Vx)$ incorporates a normalization step for $Vx$, fundamentally altering the gradient behavior. For the gradient with respect to $x$ :

$$\frac{\partial \mathcal{L}}{\partial x} \approx \frac{\alpha}{\|x\|_{RMS}} \odot \frac{\partial \mathcal{L}}{\partial y} \cdot \left( I - \frac{zz^\top}{n\|z\|_{RMS}^2} \right) \frac{V}{\|V\|_F}, \tag{33}$$

Due to the SDD network design, hidden embedding $x$ typically follows a standard normal distribution $\mathcal{N}(0,1)$. According to Theorem 3.1.1 (Vershynin, 2018), $\|x\|_{RMS}$ lies within a small neighborhood of 1, i.e., $\|x\|_{RMS} \approx 1$. For simplicity, we set $\|x\|_{RMS} = 1$ by default. Hence, the gradient norm becomes:

$$\|\frac{\partial \mathcal{L}}{\partial x}\|_{RMS} \approx \|\frac{\partial \mathcal{L}}{\partial y}\|_{RMS}. \tag{34}$$

This equality implies that the gradient magnitude is preserved during backpropagation, neither exploding nor vanishing. The combination of normalization and initialization ensures that the network maintains stable gradients, regardless of the depth or dimensionality of the layers. □

## F ARCHITECTURAL CONFIGURATION

Table 3 presents the architectural specifications of the evaluated models, including the OLMo2-1B and OLMo2-581M dense models, as well as the OLMoE-588M-3B Mixture of Experts (MoE) model. Key attributes such as parameter counts, hidden dimensions, attention configurations, and expert routing details are provided for comparison.

Table 3: Architectural Configurations of the Dense Model and MoE Model.

| Property | OLMo2-581M | OLMo2-1B | OLMo2-7B | OLMoE-588M-3B |
|---|---|---|---|---|
| Activate Params | 581M | 1B | 7B | 588M |
| Total Params | 581M | 1B | 7B | 3.4B |
| Hidden Size | 2048 | 2048 | 4096 | 1024 |
| Intermediate Size | 8192 | 8192 | 22016 | 512 |
| GQA Groups | 8 | 8 | 8 | 1 |
| Attention Heads | 32 | 32 | 32 | 16 |
| Hidden Layers | 8 | 16 | 32 | 32 |
| Experts | – | – | – | 64 |
| Topk Experts | – | – | – | 8 |
| Context Length | 4096 | 4096 | 4096 | 4096 |
| Vocabulary Size | 100278 | 100278 | 100278 | 50280 |

## G ADDITIONAL RESULTS ON DENSE MODELS

Figure 11 presents validation loss and downstream evaluation results for dense models under different training regimes. It compares SDD-1B and OLMo2-1B trained on 2T tokens with PostNorm-1B

and DeepNorm-1B trained on 200B tokens. SDD-1B consistently achieves lower validation loss and outperforms all baselines across multiple downstream tasks, highlighting its superior convergence and generalization capabilities. These results further demonstrate the advantages of Scale-Distribution Decoupling (SDD) in stabilizing optimization and improving performance in large-scale language model training.

## H    ADDITIONAL RESULTS ON MOE MODELS

Figure 12 presents the validation loss and downstream task performance of MoE models trained with 250B tokens, comparing SDD-588M-3B and OLMoE-588M-3B. SDD-588M-3B consistently achieves lower validation loss, indicating improved training stability and efficiency. Additionally, it outperforms OLMoE-588M-3B across multiple benchmarks, demonstrating superior generalization. These results highlight the benefits of Scale-Distribution Decoupling (SDD) in enhancing MoE model optimization, leading to more stable convergence and improved downstream task performance.

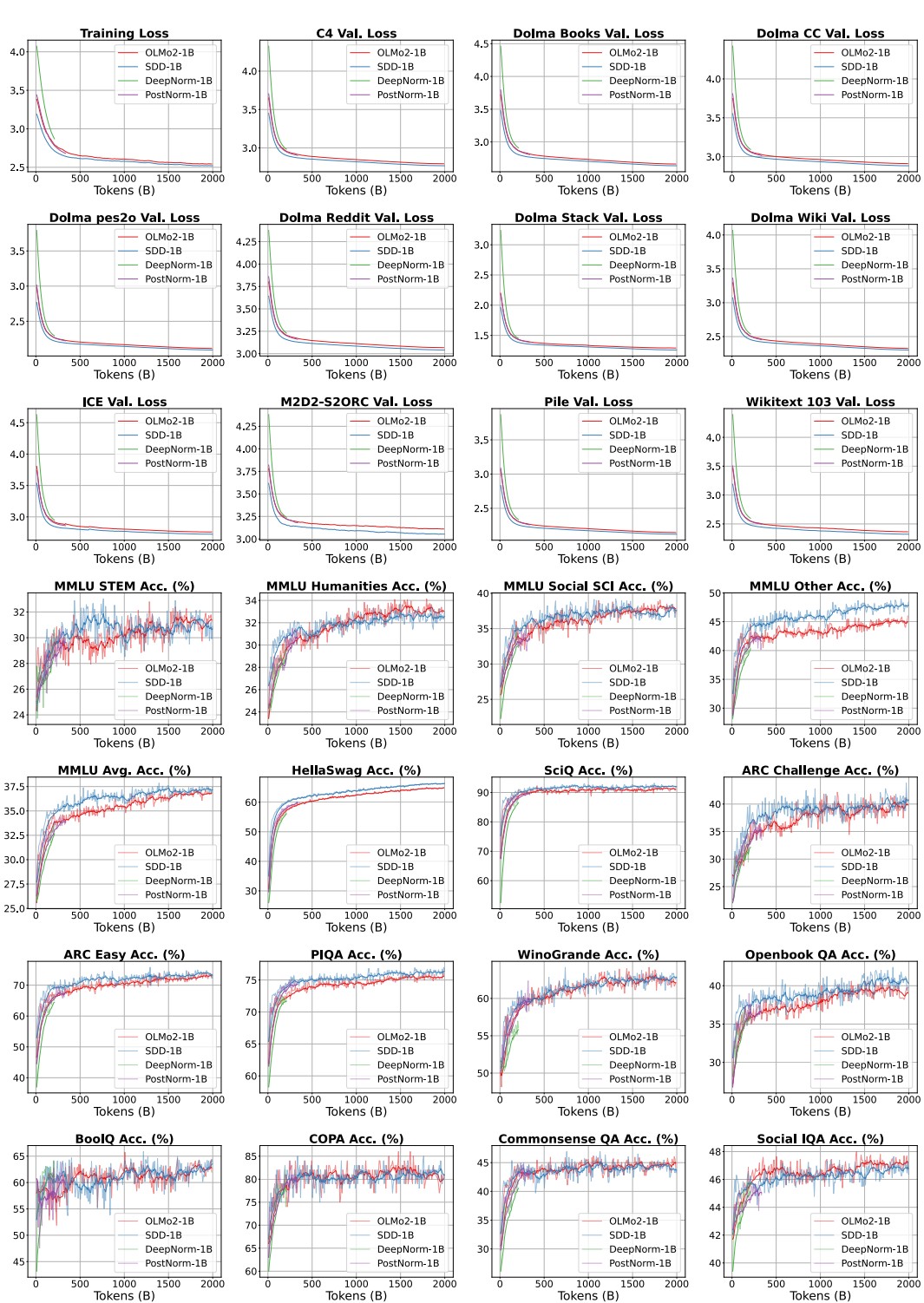

Figure 11: Training and Downstream Performance of 1B Dense Models. This figure compares validation loss and downstream task performance for SDD-1B and OLMo2-1B trained on 2T tokens, alongside PostNorm-1B and DeepNorm-1B trained on 200B tokens. SDD-1B exhibits lower loss and superior generalization, demonstrating its effectiveness in large-scale training.

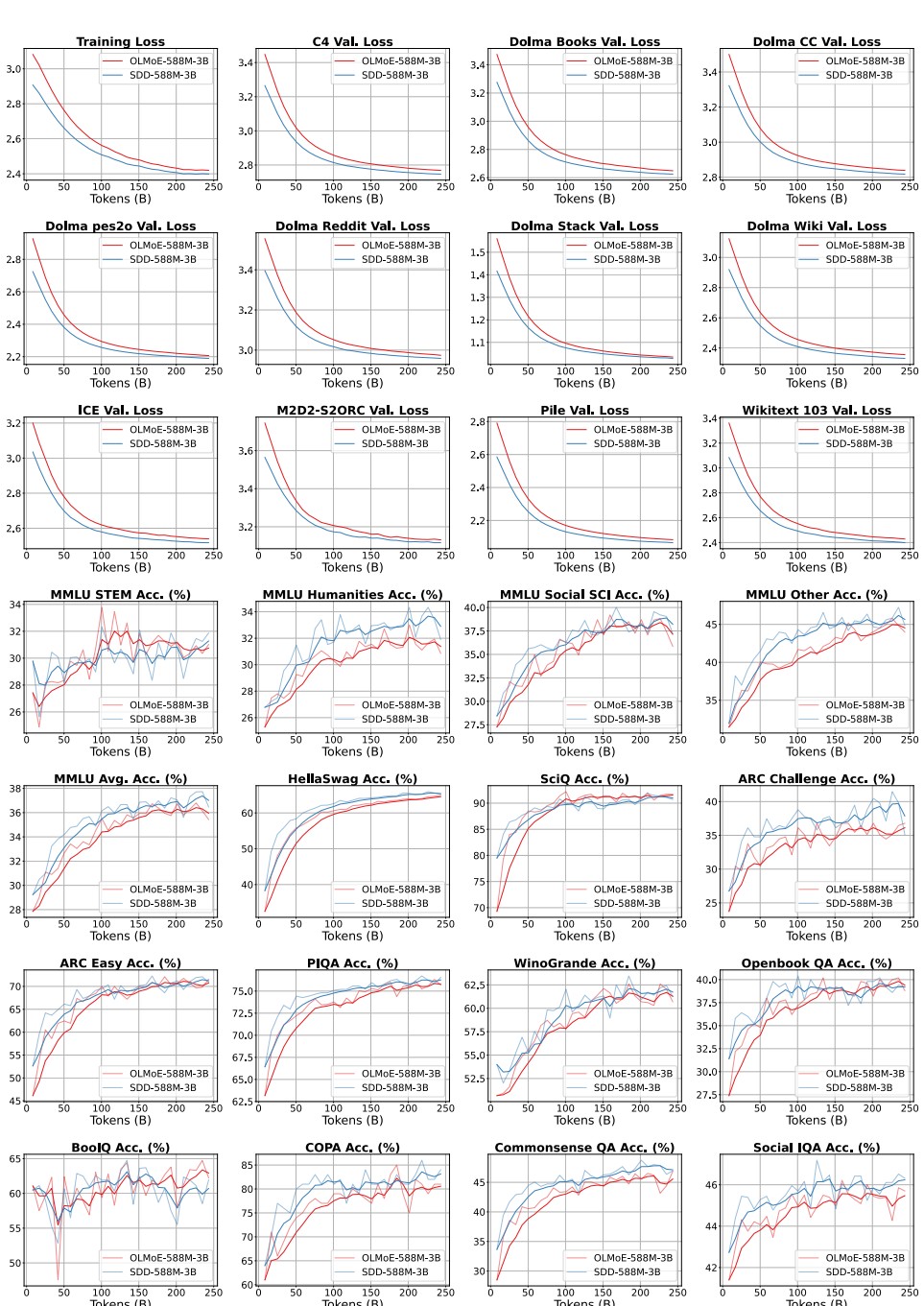

Figure 12: Training and Downstream Performance of MoE Models with 250B Tokens. This figure compares the validation loss and downstream task performance of SDD-588M-3B and OLMoE-588M-3B. SDD-588M-3B demonstrates lower loss and superior generalization across benchmarks, highlighting its effectiveness in MoE training.

