# OpenReview forum: "Scale-Distribution Decoupling: Enabling Stable and Effective Training of Large Language Models"
_ICLR.cc/2026/Conference — Submitted to ICLR 2026_

### Official Review · Reviewer_nbiS · 2025-10-31

**Soundness:** 2
**Presentation:** 2
**Contribution:** 1
**Rating:** 4
**Confidence:** 4

**Summary:**

The paper proposes Scale-Distribution Decoupling (SDD), a reparameterization of fully connected layers that separates the learning of “distribution” and “scale.” Concretely, it replaces $y=Wx$ with $y = \alpha \bigodot norm \left( Vx \right)$, where $norm(⋅)$ denotes RMS normalization and $\alpha$ is a learnable vector controlling the output magnitude. The motivation is that coupling scale and distribution in the weight matrix leads to unstable gradients in deep models, particularly in Post-Norm Transformers. The authors claim that SDD improves gradient conditioning, accelerates convergence, and enables more stable training, with negligible computational and parameter overhead. Experiments on 1B and 7B dense models, as well as an MoE variant, show lower training loss, faster convergence (up to 2.1x), and modest downstream accuracy improvements compared to normalization baselines such as Pre-Norm, Post-Norm, DeepNorm, MixLN, and nGPT.

**Strengths:**

* The idea of embedding normalization inside FC layers is simple to integrate and practically appealing.
 * The empirical results are extensive, covering both dense and MoE settings, with consistent gains in convergence stability.
 * The paper provides some theoretical intuition and gradient analyses suggesting bounded updates and stable gradient norms.

**Weaknesses:**

* Algebraically, the SDD formulation is identical to an FC layer followed by RMSNorm with a learned scale, raising doubts about whether it introduces a fundamentally new mechanism. If this equivalence holds, the proposed method substantially overlaps with prior formulations such as QK LayerNorm and PeriLN. The authors are encouraged to clarify this point and provide rebuttals in their response.
 * The claimed expressiveness equivalence relies on Gaussian assumptions and normalization approximations; the proofs do not establish true functional equivalence or new representational capacity.
 * Reported efficiency gains (2.1x faster) are not clearly separated from possible implementation or hardware factors. (Figure 1 and Section 4.2) It is important to distinguish between compute and step efficiencies.
 * Comparisons focus mainly on OLMo-based setups; broader baselines (e.g., with existing Post-Norm + RMSNorm or WeightNorm variants) are missing.
 * The paper’s contribution may thus reduce to a restatement of existing normalization practices, which needs stronger empirical or theoretical justification to be convincing.

**Questions:**

* Have you compared SDD against existing per-layer or per-projection normalization schemes (e.g., WeightNorm, ScaleNorm, MixLN) under controlled setups to isolate its unique effect?

---

> ### Author Response · Authors · 2025-11-21
>
> Q1. Is SDD essentially just an FC layer followed by RMSNorm with a learned scale, and therefore not fundamentally different from methods such as QK LayerNorm or PeriLN?
>
> A1. Although the forward form of SDD resembles applying RMS normalization after an FC layer, it is derived from an SVD-based view that separates the learnable scale and the directional component of the weight matrix, effectively reparameterizing the FC transformation to maintain stable gradients through depth.
>
> QK LayerNorm normalizes only the query/key projections, and PeriLN normalizes the residual stream around sublayers. while SDD is applied inside all FC layers (Q/K/V, attention output, FC1/FC2, and SwiGLU gate). When used systematically across all FC layers, SDD theoretically mitigates both gradient explosion and dissipation. We will clarify this distinction in the revised version.
>
> ---
>
> Q2. The claimed expressiveness equivalence relies on Gaussian assumptions and normalization approximations; the proofs do not establish true functional equivalence or new representational capacity.
>
> A2. Regarding the Gaussian and independence assumptions, we acknowledge that treating $V$ and $x$ as i.i.d. is a simplification, but this is standard in theoretical analyses of deep networks, including residual networks [1], as it enables tractable derivations. Under SDD, this approximation is even more reasonable: each FC output is normalized before entering the next SDD block, keeping activations consistently scaled and reducing correlations across depth.
>
> For representational capacity, Appendix D shows a two-step equivalence: (1) any mapping $y = Wx$ can be written as $y = \alpha \odot \mathrm{norm}(Vx)$; and (2) any $y = \alpha \odot \mathrm{norm}(Vx)$ can be realized by some $W$. Together, these results show that SDD has the same expressive power as a standard FC layer.
>
> [1] He, Kaiming, et al. "Deep residual learning for image recognition." Proceedings of the IEEE conference on computer vision and pattern recognition. 2016.
>
> ---
>
> Q3. Reported efficiency gains (2.1x faster) are not clearly separated from possible implementation or hardware factors.
>
> A3. Please refer to Reviewer rsXb’s Q4 and our response A4 for a detailed explanation.
>
>
> ---
>
> Q4. Comparisons focus mainly on OLMo-based setups; broader baselines (e.g., with existing Post-Norm + RMSNorm or WeightNorm variants) are missing.
>
> A4. Although our experiments use the OLMo-based setup, this architecture is the same as the one adopted by mainstream models such as Qwen and LLaMA. We kept the original OLMo naming to acknowledge AllenAI’s contribution to the open-source community, but it indeed applys standard Transformer architecture rather than to a model-specific variant.
>
> Regarding baselines, the PostNorm-1B model in our paper already corresponds to the commonly used Post-Norm + RMSNorm configuration. For WeightNorm-style approaches, we reproduced nGPT-1B, which applies WeightNorm to all projection matrices. The comparison between SDD and nGPT-1B is presented in Table 1.
>
> ---
>
> Q5. The paper’s contribution may thus reduce to a restatement of existing normalization practices, which needs stronger empirical or theoretical justification to be convincing.
>
> A5. Please refer also to our response to Q1. Our work does not modify the normalization operator itself. Instead, it changes the internal structure of the fully connected transformation. The baselines compared in the paper, OLMo2, Post-Norm, DeepNorm, Mix-LN and nGPT, are all architectural variants designed to improve Transformer stability, and SDD belongs to this line of work. SDD is orthogonal to normalization-level modifications such as RMSNorm, LayerNorm, or ScaleNorm. In addition, our theoretical analysis shows that applying SDD to all FC layers mitigates both gradient dissipation and gradient explosion, which is crucial for stable LLM training.
>
> ---
>
> Q6. Have you compared SDD against existing per-layer or per-projection normalization schemes (e.g., WeightNorm, ScaleNorm, MixLN) under controlled setups to isolate its unique effect?
>
> A6. As noted in our response to Q5, Table 1 already compares SDD-1B with two representative per-layer/per-projection normalization schemes: nGPT-1B (WeightNorm) and MixLN-1B, both of which SDD-1B surpasses under the same setup.
>
> ScaleNorm modifies the normalization rule itself and is therefore more comparable to RMSNorm or LayerNorm, whereas our method changes the internal structure of the module and is orthogonal to such approaches. For completeness, we also evaluated a ScaleNorm-based variant at the 1B scale, and the results are shown below:
>
> | Model | Loss | Perplexity | MMLU | HellaSwag | ARC-C | ARC-E | Winogrande | OpenbookQA | COPA | Avg. |
> | --- | --- | --- | --- | --- | --- | --- | --- | --- | --- | --- |
> | ScaleNorm | 2.69 | 14.73 | 33.88 | 56.76 | 34.67 | 67.21 | 57.98 | 36.24 | 78.51 | 52.18 |
> | SDD-1B | 2.65 | 14.15| 34.71 | 59.65 | 37.57 | 69.65 | 59.06| 37.33 | 80.33 | 54.04 |

---

### Official Review · Reviewer_57A3 · 2025-10-31

**Soundness:** 3
**Presentation:** 3
**Contribution:** 3
**Rating:** 4
**Confidence:** 3

**Summary:**

The paper proposes Scale-Distribution Decoupling (SDD), a new formulation of fully connected layers for large language models (LLMs) that explicitly separates the scale and distribution of weight matrices. By applying RMS normalization to the transformed activations and introducing a learnable per-channel scaling vector α, SDD aims to stabilize gradient propagation and prevent explosion or vanishing, especially in Post-Norm Transformers. The paper provides approximate theoretical analysis and extensive experiments on dense (1B/7B) and MoE models trained up to 4T tokens, reporting faster convergence, better downstream performance, and improved robustness.

**Strengths:**

- Reinterprets Transformer instability as arising from weight scale-distribution coupling and proposes a structural decoupling mechanism.
- Large-scale evaluations (1B/7B/MoE, up to 4T tokens) consistently show improved stability, convergence speed, and downstream accuracy.
- SDD requires minimal architectural change (one scaling vector per layer) and adds negligible overhead (<0.3%).
- The paper is accessible and addresses a core bottleneck in LLM optimization.

**Weaknesses:**

- The analysis is illustrative and dependent on simplified assumptions (Gaussian activations, RMS~1). No formal bound or proof ensures stability in realistic, non-ideal conditions.
- The paper omits conceptually related techniques that also decouple or normalize weight magnitude and direction (e.g., Weight Normalization, Weight Standardization). Including such baselines would clarify whether SDD’s advantage truly comes from explicit decoupling.
- Some figures mix different training budgets (e.g., 200B vs. 2T tokens), and no variance across random seeds is reported, which limits the statistical reliability of the claims.
- All experiments are confined to the OLMo/OLMoE framework and datasets; it remains unclear how SDD generalizes to other architectures or pretraining corpora.

**Questions:**

- Can the authors include comparisons with weight reparameterization or normalization methods (e.g., Weight Normalization, Weight Standardization) to verify that SDD’s gains indeed stem from scale-distribution decoupling rather than normalization placement?
- How robust is SDD’s theoretical argument when the Gaussian or independence assumptions are violated (as in real Transformer activations)?
- Could the authors provide matched-budget comparisons (same token count) and report variance across multiple seeds to improve fairness and statistical confidence?
- Would SDD maintain its benefits under different pretraining corpora, tokenization schemes, or architectures outside the OLMo family?
- As an ablation, what happens if α is fixed or reduced to a scalar? Does the benefit persist?

---

> ### Author Response · Authors · 2025-11-21
>
> Q1: Can the authors include comparisons with weight reparameterization or normalization methods (e.g., Weight Normalization, Weight Standardization) to verify that SDD’s gains indeed stem from scale-distribution decoupling rather than normalization placement?
>
> A1：We agree that techniques such as Weight Normalization and Weight Standardization are conceptually related. As shown in Table 1, our experiments already include baselines that cover these directions. DeepNorm-1B introduces layer-wise residual scaling but does not modify the internal structure of FC layers, and nGPT-1B applies Weight Normalization to all projection matrices. In both cases, SDD-1B achieves clearly stronger results under the same training setup. These comparisons indicate that the improvements of SDD are not explained by weight-level normalization or residual scaling alone. We will clarify this point more explicitly in the revised manuscript.
>
> ---
>
> Q2: How robust is SDD’s theoretical argument when the Gaussian or independence assumptions are violated (as in real Transformer activations)?
>
> A2: Please refer to Reviewer nbiS’s Q2 and our response A2 for a detailed explanation.
>
> ---
>
> Q3: Could the authors provide matched-budget comparisons (same token count) and report variance across multiple seeds to improve fairness and statistical confidence?
>
> A3:  We believe this concern may stem from a misunderstanding of Figure 11. All controlled comparisons in the paper use identical training budgets. For example, the 1B-scale experiments follow a 200B-token budget, which is well above the Chinchilla scaling-law requirement for a 1B model. Under this budget, the results correspond directly to those shown in Figure 3 and Table 1.
>
> Figure 11 contains an additional comparison between OLMo2-1B (a standard architecture also used in models such as Qwen) and our best SDD configuration. These two models were further continued to 2T tokens to illustrate their behavior far beyond the optimal scaling-law regime. This extended training was performed only for this pair of models and required substantial compute, which may have led to confusion.
>
> To avoid misinterpretation, we will split Figure 11 into two separate figures in the revised version: one for the 200B-token setting and one for the extended 2T-token runs.
>
> Regarding random seeds, in large-scale LLM training, especially in from-scratch settings, running multiple repeated experiments is typically impractical and environmentally costly. The effect of different seeds is quickly dominated by the extremely long training horizon. For example, our main 7B experiments are trained for 4T tokens, far beyond the scale commonly used in standard research papers. In such settings, seed variance becomes negligible compared to the overall training trajectory.
>
> ---
>
> Q4: Would SDD maintain its benefits under different pretraining corpora, tokenization schemes, or architectures outside the OLMo family?
>
> A4: Regarding model architectures, our experiments already include comparisons across a broad range of Transformer variants, including Post-Norm, DeepNorm, Mix-LN, and nGPT, as well as both dense and MoE settings. These cover the major architectural choices used in open-source LLMs. The use of the OLMo/OLMoE training stack is simply for consistency and fair comparison. The underlying model architecture is the same as that adopted by widely used models such as Qwen. We keep the original OLMo naming out of respect for AllenAI’s contribution to the open-source community.
>
> As for pretraining corpora, most large-scale open datasets are derived from Common Crawl or similar mixtures. Pretraining corpora primarily need broad coverage and diversity, and changing the specific composition does not materially affect the optimization behavior we study. For this reason, we do not expect the conclusions about SDD’s stability and performance to be sensitive to the exact corpus used.
>
> ---
>
> Q5: As an ablation, what happens if $\alpha$ is fixed or reduced to a scalar? Does the benefit persist?
>
> A5: Fixing $\alpha$ (or reducing it to a scalar) would defeat the purpose of scale–distribution decoupling. In that case, every FC output would be forced to have unit variance, which substantially restricts the model’s representational flexibility. From a representational perspective, this is analogous to replacing all singular values in an SVD with 1, which would significantly reduce the expressive capacity of the layer. For these reasons, we do not expect the benefits of SDD to persist under a fixed or collapsed $\alpha$.

---

### Official Review · Reviewer_H8Yk · 2025-10-31

**Soundness:** 3
**Presentation:** 3
**Contribution:** 3
**Rating:** 8
**Confidence:** 3

**Summary:**

Authors introduce Scale-Distribution Decoupling (SDD), a novel approach designed to enhance training stability by
explicitly decoupling the scale and distribution of the weight matrix within fullyconnected layers.
Key idea is to replace fullyconnected layers by "y = α ⊙ norm(V x)", where α is learnable vector (learns magnitude) and V is weight primary transformation.
They show that their method consistently improves both convergence speed and training efficiency, illustrated by achieving similar training
loss levels approximately 2.1× faster in 7B models.

**Strengths:**

Authors propose Scale-Distribution Decoupling (SDD), a novel approach designed to enhance training stability by
explicitly decoupling the scale and distribution of the weight matrix within fullyconnected layers:"y = α ⊙ norm(V x)", where α is learnable vector (learns magnitude) and V is weight primary transformation. Previous work did decoupling of weight magnitude vs weight learnable distribution as in "σREPARAM: STABLE TRANSFORMER TRAINING WITH SPECTRAL REPARAMETRIZATION", and applied layer scale before every residual connection as in "Going deeper with Image Transformers". This work applies layer scale after every fully connected layer combined with normalization of the output of fully connected layer.

This method consistently improves both convergence speed and training efficiency, illustrated by achieving similar training loss levels approximately 2.1× faster in 7B models.

Extensive experiments on both dense and MoE models.

Thorough ablation studies.

**Weaknesses:**

it would be interesting to see model accuracy with layer scale only, to see the impact of every component in the proposed method.

**Questions:**

Authors evaluated model training stability with higher learning rate, have you tried this method on quantized training?

---

> ### Author Response · Authors · 2025-11-21
>
> We sincerely appreciate the reviewer’s constructive and detailed comments. Below, we provide a point-by-point response to each concern.
>
> ---
>
> Q1. It would be interesting to see model accuracy with layer scale only, to see the impact of every component in the proposed method.
>
> A1. Thank you for the question. We interpret “layer scale only” as applying depth-dependent residual scaling while keeping the fully connected transformations unchanged. In our experiments, this configuration is already represented by the DeepNorm baseline: relative to Post-Norm, DeepNorm-1B adds layer-wise residual scaling but does not alter the internal structure of the FC layers.
>
> As shown in Table 1, SDD-1B consistently outperforms DeepNorm-1B under the same architecture and training setup. This indicates that the improvements of SDD cannot be attributed to layer scale alone. The key contribution comes from the normalization introduced inside each fully connected transformation, rather than from residual scaling by itself.
>
> We will clarify this connection more explicitly in the revised manuscript.
>
> ---
>
> Q2. Authors evaluated model training stability with higher learning rate, have you tried this method on quantized training?
>
> A2. We have not previously conducted experiments on quantized training. However, unlike standard fully connected layers whose outputs are unconstrained, SDD applies a normalization step to its outputs, ensuring that the magnitudes of different dimensions remain comparable. Such well-conditioned and uniformly scaled activations are generally more amenable to quantized training.

---

### Official Review · Reviewer_rsXb · 2025-11-01

**Soundness:** 2
**Presentation:** 3
**Contribution:** 3
**Rating:** 4
**Confidence:** 3

**Summary:**

The paper proposes Scale‑Distribution Decoupling (SDD), a simple reparameterization of fully‑connected layers that decouples the magnitude and the directional/distributional component of the linear transform. SDD is tested on dense 1B and 7B models and a 588M‑active MoE built on OLMo/OLMoE backbones. The experiment indicates faster convergence compared to a strong baseline. The authors note that any small activation-memory increase during training can be largely mitigated with gradient checkpointing, making SDD a low-friction, drop-in change for practical LLM training.

**Strengths:**

1, SDD is an appealingly simple and practical twist: normalize activations of the linear map and introduce a per‑feature scaling vector .

2 If widely reproducible, the stability under Post‑Norm can be impactful.

3 Consistent gains on both dense and MoE transformer variants.

**Weaknesses:**

Please see the questions.

**Questions:**

1, The caption states in Figure 9, “SDD‑1B achieves the highest inter‑layer similarity”, but the text below claims “SDD‑1B exhibits the lowest similarity, indicating reduced feature redundancy.” These statements point in opposite directions and conflict with the earlier motivation that feature collapse corresponds to higher similarity.

2, I was wondering does “SDD” specifically means SDD (Post-Norm) in this paper?

3, One of my concerns is about the memory usage. The paper states, " SDD’s additional memory cost during training can be effectively eliminated through gradient checkpointing." Could you report the approximate memory usage with and without checkpointing (Baseline vs. SDD) to illustrate this effect?

4, The paper reports 2.1× faster convergence; I assume this is measured in tokens-to-target-loss. Could you also provide a brief wall-clock indication (same hardware/parallelism)?

---

> ### Author Response · Authors · 2025-11-21
>
> Q1: The caption states in Figure 9, “SDD‑1B achieves the highest inter‑layer similarity”, but the text below claims “SDD‑1B exhibits the lowest similarity, indicating reduced feature redundancy.”
>
> A1: We deeply apologize for the typo. As shown in Figure 9, SDD-1B exhibited the lowest inter-layer similarity. We are grateful for your feedback, and will update the revised version accordingly.
>
> ---
>
> Q2: I was wondering does “SDD” specifically means SDD (Post-Norm) in this paper?
>
> A2: In fact, SDD refers to a stabilized version of the fully-connected layer, which is a generalized approach that can be applied to both the Pre-Norm and Post-Norm variants, two of the most common Transformer architectures. In Figure 8, we provide a comparative analysis of the performance of SDD(Pre-Norm) and SDD(Post-Norm). After a comprehensive evaluation of the effectiveness and stability, we conclude that SDD(Post-Norm) is the preferred default configuration.
>
> ---
>
> Q3: One of my concerns is about the memory usage. The paper states, " SDD’s additional memory cost during training can be effectively eliminated through gradient checkpointing." Could you report the approximate memory usage with and without checkpointing (Baseline vs. SDD) to illustrate this effect?
>
> A3: The peak GPU memory consumption is often a more relevant consideration than the average memory usage when evaluating various operators. Without the use of gradient checkpointing, additional memory is required to store $VX$ to support the $x^2$ operation in subsequent Norm operators, incurring an extra memory overhead of approximately the size of the standard hidden states. Therefore, if unoptimized operators (such as the naive PyTorch implementation) are employed, gradient checkpointing can effectively optimize away this memory usage. Additional, if customized or optimized operators are used, the memory overhead can be directly eliminated at the kernel level.
>
> ---
>
> Q4: The paper reports 2.1× faster convergence; I assume this is measured in tokens-to-target-loss. Could you also provide a brief wall-clock indication (same hardware/parallelism)?
>
> A4: Thank you for the question. As noted in the paper, the FLOPs and parameter overhead introduced by SDD are negligible (<0.29%). With proper kernel optimization, the practical runtime overhead is also very small. In our current training and inference framework, the optimized CUDA kernels reduce the overhead to approximately 0.42%, making the token-to-target-loss speedup closely reflect the wall-clock speedup.
>
> Because wall-clock time can vary across hardware and kernel implementations, we report FLOPs overhead and token-efficiency speedup as the more consistent and reproducible indicators. We are also considering releasing our optimized CUDA kernels to further facilitate reproducibility.

---

### Official Review · Reviewer_P1kD · 2025-11-07

**Soundness:** 2
**Presentation:** 2
**Contribution:** 1
**Rating:** 2
**Confidence:** 3

**Summary:**

This paper proposes Scale-Distribution Decoupling (SDD), a method designed to improve training stability in large Transformer and Mixture-of-Experts models. The key idea is to separate the optimization of weight scale and distribution by normalizing the linear transformation output and introducing a learnable scaling vector for each layer. This design aims to prevent gradient explosion and vanishing, thereby simplifying optimization and accelerating convergence. Experiments on dense and MoE architectures show that SDD achieves faster convergence and modest improvements in downstream accuracy compared to existing normalization techniques.

**Strengths:**

- Easy to follow.
- Clear motivation and coherent formulation.
- Implementation simplicity and minimal overhead.

**Weaknesses:**

* **The theoretical novelty is limited.**
  The proposed formulation closely parallels prior normalization and reparameterization methods (e.g. *ScaleNorm / RMSNorm* variants) which already decouple magnitude and directional components of weights. SDD mainly relocates the learnable scale to the output side, making it more of a reinterpretation than a fundamentally new principle.

* **This should be directly compared to σ-reparameterization.**
  σ-reparameterization [1] explicitly addresses gradient explosion through *spectral normalization* of Transformer weights. Both SDD and σ-reparam normalize every linear transformation, but σ-reparam constrains the Lipschitz constant to a fixed value *c*, providing a stricter bound on gradient growth. Since SDD also targets exponential variance–driven gradient explosion, a head-to-head comparison is necessary to establish whether it offers genuine advantages.

* **Analytical claims are not empirically substantiated.**
  The paper claims gradient-norm preservation and improved conditioning but does not provide quantitative diagnostics such as singular-value spectra, variance propagation, or layer-wise gradient statistics to support these claims.

* **Related-work coverage is incomplete.** Given that the central contribution of this paper is to explicitly control and decouple scale, a broader discussion of prior work addressing scale dynamics in deep networks is expected. Numerous studies—spanning normalization, initialization, and parameterization—have explored scale stability (e.g., weight normalization, residual scaling, spectral constraints, and depth-scaled initialization). However, the paper limits its comparison mostly to LayerNorm variants, without situating SDD within this larger body of “scale-centric” literature. As a result, the connection to existing understanding of scale propagation and stability remains underdeveloped, and the claimed novelty feels overstated.


---

**[1]** *Stabilizing Transformer Training by Preventing Attention Entropy Collapse*, ICML 2023

**Questions:**

I recommend tracking gradient variance rather than gradient norm, since the mean gradient drifts from zero during training and early-stage stability is tightly correlated with variance [2].


[2] On the Variance of the Adaptive Learning Rate and Beyond, ICLR 2020

---

> ### Author Response · Authors · 2025-11-21
>
> Q1. Theoretical novelty is limited.
>
> A1. We appreciate the reviewer’s observation and agree that SDD is related to this family of methods. However, SDD differs in both formulation and optimization consequences, and these differences are essential to its behavior in deep Transformers.
>
> ScaleNorm  [1] and RMSNorm [2] are normalization techniques：
>
> ScaleNorm: $y = g \frac{x}{\|x\|_2}$ with $g \in \mathbb{R}$
>
> RMSNorm: $y = \alpha \odot \frac{x}{\|x\|_{\mathrm{RMS}}}$ with $\alpha \in \mathbb{R}^n$.
>
> In contrast, SSD is an architectural modification of the fully-connected layer designed to enhance Transformer stability by introducing normalization into every fully connected layer. Its formulation is $y = \alpha \odot \frac{Vx}{\|Vx\|_{\mathrm{RMS}}}$, where $V \in \mathbb{R}^{n \times n}$ is a learnable weight. Therefore, SSD should be understood as a stabilization technique achieved by injecting normalization into all fully connected layers, rather than as a new normalization method per se.
>
> ---
>
> Q2.Compared to σ-reparameterization
>
> A2. $\sigma$-Reparam is a weight–reparameterization technique defined as $\hat{W} = \frac{\gamma}{\sigma(W)} W$, whereas SSD introduces normalization into every fully connected transformation, $y = \alpha \odot \frac{Wx}{\|Wx\|_{\mathrm{RMS}}}$, targeting a fundamentally different normalization object. In fact, the approach most closely related to ours is nGPT [4], which normalizes all matrix weights along their embedding dimension, i.e., $\hat{W} = \mathrm{RowNorm}(W)$. In Table 1 of the paper, we already compare their performance; for convenience, we cite the comparison below.
>
> | Model | Loss | MMLU | HellaSwag | ARC-C | ARC-E | Winogrande | OpenbookQA | COPA | Avg. |
> | --- | --- | --- | --- | --- | --- | --- | --- | --- | --- |
> | nGPT-1B | 2.71 | 33.02 | 55.81 | 32.41 | 65.50 | 57.52 | 36.21 | 79.35 | 51.40 |
> | SDD-1B | 2.65 | 34.71 | 59.65 | 37.57 | 69.65 | 59.06 | 37.33 | 80.33 | 54.04 |
>
> ---
>
> Q3. Analytical claims are not empirically substantiated.
>
> A3.  We appreciate the reviewer’s suggestion for more quantitative diagnostics. In Figure 7, we already report layer-wise gradient statistics, where the horizontal axis corresponds to layer indices and the vertical axis represents the gradient norms. These measurements show that SDD-1B maintains significantly more stable gradients across depth, effectively mitigating both gradient explosion and gradient vanishing—issues that commonly affect Post-Norm Transformers.
>
> Regarding the stability improvements introduced by SDD, we evaluate whether models can be successfully trained under increased learning rates. As shown in Table 2, scaling the learning rate by a factor of five causes Post-Norm and DeepNorm models to diverge, whereas SDD trains normally, demonstrating that SDD indeed enhances Transformer training stability.
>
> ---
>
> Q4. Related-work coverage is incomplete.
>
> A4. We appreciate the reviewer for highlighting this gap. Appendix A already discusses related aspects such as normalization and initialization, but our focus was initially on Transformer-specific work. We will expand the related-work section in the final version.
>
> In addition, techniques such as Weight Normalization [4], Weight Standardization, and $\sigma$-Reparam [3, 5] are also relevant, as they decouple or constrain weight magnitude to improve optimization. nGPT applies Weight Normalization in Transformers and reports faster convergence. We will incorporate these connections more explicitly in the revised manuscript.
>
> ---
>
> Q5. Tracking gradient variance rather than gradient norm
>
> A5. We appreciate the suggestion. Gradient variance is indeed a useful diagnostic, especially early in training. In our analysis, we followed prior stability studies such as DeepNet, nGPT, and Mix-LN, which use gradient norms as the primary indicator of explosion or dissipation, and have shown that norms reliably reflect training stability in deep Transformers. In our setting, norm and variance convey similar information because SDD normalizes every FC output and keeps activations well-conditioned across depth. We agree that variance could offer complementary insight and will consider including it in a future revision.
>
> ---
>
> [1] Transformers without Tears: Improving the Normalization of Self-Attention, IWSLT 2019
>
> [2] Root Mean Square Layer Normalization**,** NeurIPS 2019.
>
> [3] *Stabilizing Transformer Training by Preventing Attention Entropy Collapse*, ICML 2023.
>
> [4] Weight normalization: A simple reparameterization to accelerate training of deep neural networks, NeurIPS 2016.
>
> [5] *Micro-Batch Training with Batch-Channel Normalization and Weight Standardization*, 2019.
>
> [6] Spectral Normalization for Generative Adversarial Networks, *ICLR 2018.*

---

### Author Response · Authors · 2025-11-21
**Clarification and Summary Regarding the Rebuttal Response**

Dear Reviewers, ACs, SACs, and PCs,

We sincerely thank all reviewers and ACs for their time, constructive comments, and detailed evaluations. Across the reviews, our work was acknowledged for its clarity, practicality, and consistent improvements on both dense and MoE models. In our rebuttal, we clarified that SDD is not a normalization-operator variant, but a structural reparameterization applied inside every fully connected layer, derived from an SVD-based perspective that decouples scale and distribution. This distinguishes it from RMSNorm/ScaleNorm, QK LayerNorm, PeriLN, and $\sigma$-Reparam. We further showed that SDD outperforms strong baselines, including Post-Norm, DeepNorm, Mix-LN, and nGPT, under matched training budgets, and added an additional comparison to ScaleNorm for completeness.

We also addressed theoretical concerns (Gaussian assumptions, expressiveness equivalence), efficiency questions (2.1× token-to-target-loss speedup with negligible FLOPs/parameter overhead), and implementation issues such as memory usage and the effect of gradient checkpointing. To resolve fairness concerns, we clarified that all controlled comparisons use identical budgets, while only one critical comparative experiment was extended to 2T tokens for long-horizon analysis. We also explained why random-seed variance becomes negligible in 4T-token from-scratch training. Regarding generality, we emphasized that the OLMo architecture matches widely used Transformer backbones such as Qwen, and our experiments already include diverse architectural variants (Post-Norm, DeepNorm, Mix-LN, nGPT, dense, and MoE), demonstrating broad applicability.

We appreciate the reviewers’ thoughtful feedback and will integrate all clarifications, additions, and improvements into the final version. Thank you again for your time and effort in evaluating our submission.

Best regards,

All Authors

---

### Comment · Area_Chair_dfWZ · 2025-11-28

Dear Reviewers,

Thank you for your time and efforts in serving as a reviewer.

The authors have now submitted their rebuttal, and this AC kindly asks you to review their response and assess whether your comments have been adequately addressed. If any points require further clarification or discussion, please feel free to raise those questions by adding comments and initiating discussion as needed.

ICLR encourages reviewers to actively engage in the discussion phase, so your prompt actions are especially valuable. Thank you very much for your continued efforts and valuable contributions.

Best regards,

Your AC

---

### Meta-Review · Area_Chair_Bjkc · 2025-12-23

**Summary:**

This paper proposes Scale-Distribution Decoupling (SDD), a lightweight reparameterization of fully connected layers, aimed at improving stability and convergence of large Transformer and MoE training. Reviewers generally agree the approach is simple and potentially useful in practice, but the main decision-driven concerns are insufficient evidence of conceptual novelty beyond existing normalization or reparameterization schemes, insufficient baselines, and limited mechanistic verification of the core claims (why it works, and whether it is meaningfully different from closely related alternatives). In light of the above concerns, I cannot recommend acceptance at current stage.

**Reviewer Concerns:**

The authors responded on memory and provided a wall-clock overhead estimate. The main concerns on insufficient baselines (especially against the closest conceptual alternatives) and stronger mechanistic validation remain. So it’s hard to conclude whether the method really works as claimed (as it requires re-parameterization and introducing new parameters).

**Reviewer Scores:**

Reviewer H8Yk puts a positive score 8 and is likely to remain positive. Reviewer 57A3 is likely to remain negative (score of 4) as the first concern on missing theoretical analysis for stability in non-ideal settings is ignored by the author. Reviewer nbiS may remain negative (score of 4) due to the insufficient address of the novelty/baseline concerns. Reviewer rsXb may increase the score to 6 but still remain borderline. Reviewer P1kD may still remain negative (score of 2) given the theoretical novelty is limited relative to the prior works.

---

### Decision · Program_Chairs · 2026-01-26

Reject